# Applying the cost of generating force hypothesis to uphill running

Wouter Hoogkamer[1,2], Paolo Taboga[3] and Rodger Kram[3]

[1] Faculty of Human Movement Sciences, VU University, Amsterdam, The Netherlands
[2] Department of Kinesiology, KU Leuven, Leuven, Belgium
[3] Department of Integrative Physiology, University of Colorado Boulder, CO, USA

Corresponding author
Wouter Hoogkamer,
wouter.hoogkamer@faber.kuleuven.be

## ABSTRACT

Historically, several different approaches have been applied to explain the metabolic cost of uphill human running. Most of these approaches result in unrealistically high values for the efficiency of performing vertical work during running uphill, or are only valid for running up steep inclines. The purpose of this study was to reexamine the metabolic cost of uphill running, based upon our understanding of level running energetics and ground reaction forces during uphill running. In contrast to the vertical efficiency approach, we propose that during incline running at a certain velocity, the forces (and hence metabolic energy) required for braking and propelling the body mass parallel to the running surface are less than during level running. Based on this idea, we propose that the metabolic rate during uphill running can be predicted by a model, which posits that (1) the metabolic cost of perpendicular bouncing remains the same as during level running, (2) the metabolic cost of running parallel to the running surface decreases with incline, (3) the delta efficiency of producing mechanical power to lift the COM vertically is constant, independent of incline and running velocity, and (4) the costs of leg and arm swing do not change with incline. To test this approach, we collected ground reaction force (GRF) data for eight runners who ran thirty 30-second trials (velocity: 2.0–3.0 m/s; incline: 0–9°). We also measured the metabolic rates of eight different runners for 17, 7-minute trials (velocity: 2.0–3.0 m/s; incline: 0–8°). During uphill running, parallel braking GRF approached zero for the 9° incline trials. Thus, we modeled the metabolic cost of parallel running as exponentially decreasing with incline. With that assumption, best-fit parameters for the metabolic rate data indicate that the efficiency of producing mechanical power to lift the center of mass vertically was independent of incline and running velocity, with a value of ∼29%. The metabolic cost of uphill running is not simply equal to the sum of the cost of level running and the cost of performing work to lift the body mass against gravity. Rather, it reflects a constant cost of perpendicular bouncing, decreased costs of parallel braking and propulsion and of course the cost of lifting body mass against gravity.

## INTRODUCTION

The energetic cost of running affects the behavior/performance of animals in nature, humans seeking fitness and athletes in competition. We believe that reasonable

biomechanical explanations for the energetic cost of level running have been developed (*Alexander & Ker, 1990*; *Arellano & Kram, 2014*; *Kram & Taylor, 1990*; *Minetti & Alexander, 1997*; *Roberts et al., 1998*), but the world is not flat. We all know intuitively that running up even a slight incline is dramatically more exhausting, yet we lack a coherent biomechanical model for the energetic cost of uphill running.

In this paper, we develop and test a new model for the metabolic cost of uphill human running. Historically, several different approaches have been applied to this topic. Most of these approaches result in unrealistically high values for the efficiency of performing vertical work (*Asmussen & Bonde-Petersen, 1974*; *Lloyd & Zacks, 1972*; *Pugh, 1971*), or are only valid for running up steep inclines (*Margaria et al., 1963*; *Margaria, 1968*; *Minetti et al., 2002*) and not for running up inclines more typical of recreational/fitness running. The purpose of the current study was to reexamine the cost of uphill running, based upon our understanding of level running energetics (*Kram & Taylor, 1990*; *Roberts et al., 1998*) and ground reaction forces during uphill running (*Gottschall & Kram, 2005*).

First, we give an overview of how the energetics of uphill running have been approached in the past. Margaria and co-workers (*Margaria et al., 1963*; *Margaria, 1968*) calculated net mechanical efficiency of uphill running as:

*Net mechanical efficiency $=$ vertical mechanical power$/$net metabolic rate* (1)

Here, the vertical mechanical power is the rate of performing work to raise the body mass ($m$) against gravity ($g$):

*vertical mechanical power $= m \cdot g \cdot sin(\theta) \cdot v$* (2)

where $\theta$ is the incline in degrees and $v$ is the running velocity parallel to the incline. Margaria obtained the net metabolic rate by subtracting the basal metabolic rate from the metabolic rate during running. In level running, at a constant velocity, upon landing the body absorbs mechanical power (performs negative work) and then generates positive power (performs positive work) but no net external mechanical power is required because the negative and positive work quantities are opposite in sign but equal in magnitude (*Cavagna, Saibene & Margaria, 1964*). *Margaria (1968)* proposed that the equal and opposite positive and negative external work can be considered to be wasted, since performing this work has a metabolic cost but does not propel the runner forward.

However, in uphill running, net positive external work and power are produced since the center of mass (COM) is raised against gravity. *Margaria et al. (1963)* hypothesized and demonstrated that on steeper inclines, the wasted external work decreases and the observed net mechanical efficiency approaches the same value as the efficiency of predominantly concentric exercise, such as cycle ergometry ($\sim$25%). It is important to note that this approach only results in such physiologically realistic efficiency values when the energetic cost of running is dominated by the work needed to raise the COM (i.e., at steep inclines) (*Minetti et al., 2002*). For running up inclines more typical of recreational/fitness running the net mechanical efficiencies calculated are much lower than the values for concentric muscle contractions (*Smith, Barclay & Loiselle, 2005*).

Another approach is to calculate "vertical efficiency" by dividing the mechanical power needed to lift the COM vertically by the difference in metabolic rates between locomotion on an incline and level locomotion at the same velocity (e.g., *Full & Tullis, 1990*; *Rubenson et al., 2006*). Published values for vertical efficiency range from 30% for red kangaroos (*Kram & Dawson, 1998*) to ~46% for humans (*Asmussen & Bonde-Petersen, 1974*; *Lloyd & Zacks, 1972*; *Pugh, 1971*), to values near 50% (walking turtles; *Zani & Kram, 2008*) or even higher (60% for mice and 66% for chimpanzees; *Taylor, Caldwell & Rowntree, 1972*). In running, these efficiency values, which are much higher than isolated muscle contraction efficiency, have been attributed to elastic energy storage and reutilization in muscle–tendon complexes (*Asmussen & Bonde-Petersen, 1974*; *Lloyd & Zacks, 1972*; *Cooke et al., 1991*). But, as emphasized by *Roberts et al. (1997)*, the increase in potential energy of the body in uphill locomotion can only be done by active concentric muscle work, since passive elastic mechanisms simply return energy stored previously in a step. Thus, these high efficiency values remain enigmatic.

Alternatively, *Minetti, Ardigò & Saibene (1994)* developed a model which assumed that the metabolic cost can be predicted completely based on measures of mechanical work. In their model, internal work (due to the kinetic energy changes of body segments relative to the body COM), positive external work and negative external work were each assumed to be performed with a separate efficiency value. *Minetti, Ardigò & Saibene*'s model (*1994*) also estimates the amount elastic energy storage and release, however the costs of muscle force production to generate tension to allow this energy storage and release is not taken into account.

Although we believe that the cost of generating force to support body weight is the major determinant of the metabolic cost of level running (for review, see *Arellano & Kram, 2014*; *Kram, 2000*), none of the models for uphill running explicitly include this cost. Briefly, the cost of generating force hypothesis posits that in running the muscles primarily act to generate tension that allows the tendons to store and return elastic energy. Muscles consume energy whenever they generate tension, regardless of whether they perform work. The cost of generating force to support body weight has been found to be inversely proportional to the foot-ground contact time, presumably because generating force more rapidly requires faster and less economical muscle fibers (*Roberts et al., 1998*).

In this study, we introduce a model for the metabolic cost of uphill running which combines the cost of generating force and the cost of performing mechanical work approaches. Our overall view is that the net metabolic cost of running is comprised of the costs of generating force to support body weight, braking and propelling body mass in the forward (parallel) direction, swinging the legs and arms and maintaining balance (*Arellano & Kram, 2011*; *Arellano & Kram, in press*; *Arellano & Kram, 2014*; *Chang & Kram, 1999*; *Farley & McMahon, 1992*; *Kram & Taylor, 1990*; *Modica & Kram, 2005*; *Moed & Kram, 2005*; *Roberts et al., 1998*; *Teunissen, Grabowski & Kram, 2007*). For level running, obviously body weight must be dynamically supported in the vertical direction, but for uphill running, we prefer to call this term the cost of "perpendicular bouncing" to emphasize that the metabolic power required to lift the COM vertically is not included in that term (Fig. 1).

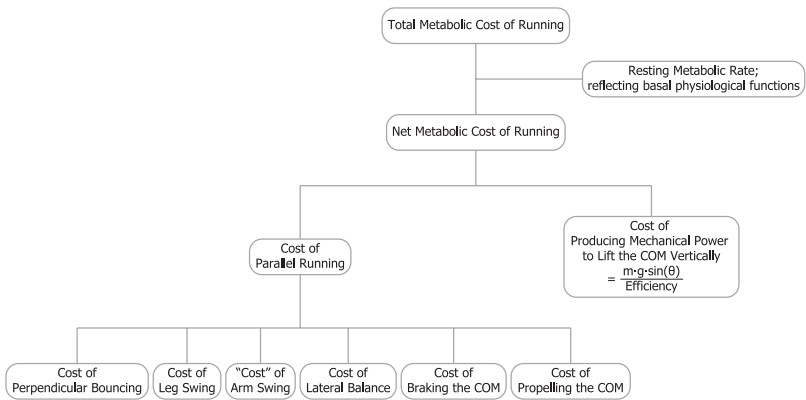

**Figure 1 The total metabolic cost of running is comprised of several components.** Parallel running refers to the task of running parallel to the surface whether that surface is level or inclined. The task of parallel running intrinsically requires bouncing perpendicular to the surface and that bouncing incurs a metabolic cost.

This approximation introduces only a small error because the perpendicular component is only slightly less than the vertical component, for example, the cosine of 9 degrees equals 0.988. Furthermore, *Gottschall & Kram (2005)* observed that both the perpendicular active force peaks and the contact times during uphill running (up 3, 6 and 9°) were not significantly different from those during level running. Thus, based on the cost of generating force hypothesis, the cost of perpendicular bouncing should not change with incline. So, in uphill running, the net metabolic rate should be equal to the sum of the rates of metabolic energy consumption for perpendicular bouncing, braking and propelling body mass parallel to the surface, swinging the legs and arms and, of course, raising of the COM vertically. In Fig. 1 parallel running refers to the task of running parallel to the surface whether that surface is level or inclined. The task of parallel running intrinsically requires bouncing perpendicular to the surface and that bouncing incurs a metabolic cost.

In contrast to the vertical efficiency approach, we propose that at a certain velocity the metabolic rate required for braking and propelling the body mass parallel to the running surface is less during inclined running (compared to level running), because there is less braking (negative external work) and thus less wasted work (*Margaria, 1968*; *Minetti, Ardigò & Saibene, 1994*). *Gottschall & Kram (2005)* quantified how in uphill running the braking Ground Reaction Forces (GRFs) parallel to the running surface decrease with steeper inclines. The propulsive GRFs parallel to the running surface are greater during uphill running, but the majority of the propulsive GRF impulse parallel to the running surface compensates for the gravitational braking impulse parallel to the surface $m \cdot g \cdot sin(\theta) \cdot t_{step}$, where $t_{step}$ is the time between two consecutive foot strikes. During steeper incline running, most of the propulsive parallel GRF impulse is required to overcome the component of the gravitational braking impulse parallel to the surface. Thus, only a small part of the parallel propulsive GRF impulse is compensating for the braking GRF impulse (Fig. 2). Although initially counterintuitive, the metabolic costs of both braking and propelling forces, parallel to the running surface, should decrease

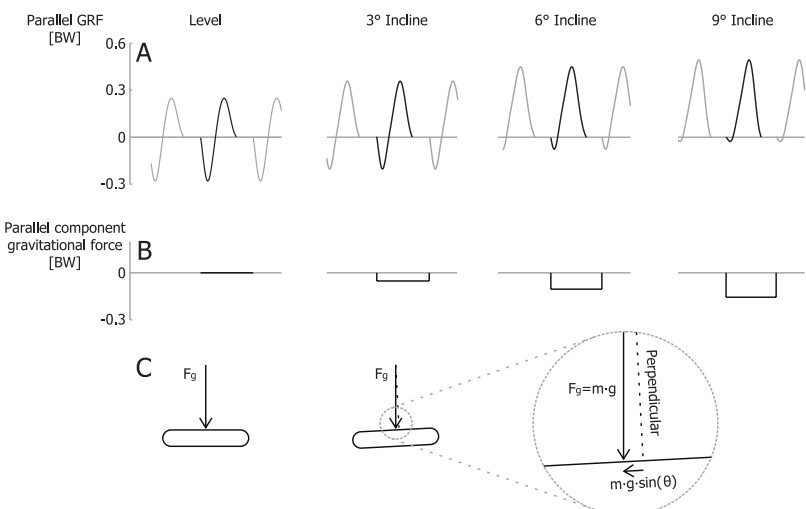

**Figure 2 Ground reaction forces for different inclines.** (A) Idealized parallel ground reaction force versus time traces for running at 3 m/s. (B) parallel component of gravitational impulse for a single step, and (C) schematic representation of the gravity force vector and its component parallel to the running surface. Forces are normalized to body weight.

during uphill running. By taking that into account, the efficiency of producing mechanical power to lift the COM vertically should be closer to the efficiency of concentric muscle contractions.

Based on these ideas, we propose that the metabolic rate during uphill running can be predicted by a model, which posits that (1) the metabolic cost of perpendicular bouncing remains the same as during level running, (2) the metabolic cost of running parallel to the running surface decreases with incline, (3) the delta efficiency (*Gaesser & Brooks, 1975*) of producing mechanical power to lift the COM vertically ($Eff_{vCOM}$) is constant, independent of incline and running velocity, and (4) the costs of leg and arm swing do not change with incline. We expect $Eff_{vCOM}$ to be similar to the delta efficiency of cycling (∼25%–30%) (*Gaesser & Brooks, 1975*; *Bijker, De Groot & Hollander, 2001*). To test these ideas, we measured GRFs for level and a range of uphill running inclines (1–9°) for a range of velocities (2.0–3.0 m/s). Additionally, we measured the metabolic rate during uphill running for a feasible range of grades at the same velocities (0–8° at 2.0 m/s; 0–4° at 2.5 and 3.0 m/s).

## MATERIALS & METHODS

### Ground reaction forces

For this part of the study, eight participants ran on a force treadmill (Treadmetrix, Park City, UT, USA). Six males and two females participated (31.2 ± 11.0 yr, 177.6 ± 7.1 cm, 69.5 ± 7.9 kg; all mean ± SD). The participants gave written informed consent and the testing protocol was approved by the University of Colorado Institutional Review Board (13-0710).

Each trial lasted 30 s and the first 20 steps after the first 10 s were analyzed. Forces were collected at 1,000 Hz. Before each trial forces were zeroed by regulating the amplifiers (MSA-6 MiniAmp, AMTI Watertown, MA, USA) and the acquisition software (Vicon Nexus, Vicon Motion Systems Ltd., Oxford, UK). Signals were digitally filtered using a first-order Butterworth filter (pass band frequency of 35 Hz and stop band frequency of 50 Hz) implemented in a Matlab script (Mathworks Inc., USA). A 10 N threshold was used to determine the instants of foot strike and toe-off. Per step, we calculated the average braking and propelling GRF impulses parallel to the running surface by integrating all negative (braking) or positive (propelling) values during each ground contact. We used the time between two consecutive foot strikes $t_{step}$ (for example from left foot strike to right foot strike) to calculate the average gravitational impulses parallel to the surface:

$$I\_Gravity_{parallel} = m \cdot g \cdot sin(\theta) \cdot t_{step} \tag{3}$$

We defined the total propelling impulse per step as the propelling GRF impulse minus the component of the gravitational impulse, all parallel to the running surface:

$$I\_GRF_{propelling\ total} = I\_GRF_{propelling} - I\_Gravity_{parallel} \tag{4}$$

Summation of the absolute values of the braking impulse and of the total propulsive impulse per step gave the value of the wasted GRF impulse per step:

$$I\_GRF_{wasted} = I\_GRF_{braking} + I\_GRF_{propelling\ total} \tag{5}$$

For level running, the wasted GRF impulse per step equals the summation of the absolute values of the braking impulse and the propulsive impulse per step, similar to the concept of wasted work per step as introduced by *Margaria (1968)*. We note that not all of the wasted impulse is actively done by muscle length changes; a substantial part is likely provided through passive elastic storage and return. Based on earlier studies with kinetic (*Gottschall & Kram, 2005*) and kinematic (*Minetti, Ardigò & Saibene, 1994*) measurements of uphill running and the fact that $I\_GRF_{wasted}$ cannot be negative, we hypothesized that $I\_GRF_{wasted}$ decreases exponentially to zero for steeper inclines:

$$I\_GRF_{wasted} = I\_GRF_{wasted\ level} \cdot e^{-\gamma \cdot sin(\theta)} \tag{6}$$

Parameter $I\_GRF_{wasted\ level}$ represents the value of the wasted GRF impulse per step during level running. The decay constant $\gamma$ determines how steeply $I\_GRF_{wasted}$ decreases with incline (expressed as $sin(\theta)$).

## Metabolic measurements

We recruited a different set of eight participants for this part of the study (4 males and 4 females, $26.2 \pm 4.0$ yr, $174.3 \pm 12.4$ cm, $67.3 \pm 11.8$ kg; all mean $\pm$ SD). All participants had extensive treadmill running experience and had recently run a 5 km race in less than 20 min ($18:28 \pm 52$ s; mean $\pm$ SD). Based on pilot testing, we were confident that for this caliber of runner, the most demanding test condition would be submaximal. We applied

this 20-minute 5 km criteria to ensure that the energy supply during our experimental trials was predominately oxidative and to avoid fatigue effects. The participants gave written informed consent that followed the guidelines of the University of Colorado Institutional Review Board (0606.29).

Participants completed different sessions on two separate days. They ran a total of 17 different conditions on a classic Quinton 18–60 treadmill with adjustable velocity and incline. Note that we modified this treadmill so that we had calibrated, digital electronic readouts for velocity and incline. On the first day, participants ran at a velocity of 2.0 m/s at seven different inclines ranging from 0 to 8°. The second day consisted of five trials at both 2.5 m/s and 3.0 m/s at inclines ranging from 0 to 4° (for a complete list of the trials, refer to Table 2 in the Results section). We measured the rates of oxygen consumption and carbon dioxide production during these 7-minute trials. Each experimental day started with determining the body mass of the participant. We then determined metabolic rate during a 7-minute standing trial. This was followed by a 10-minute warm-up of level running at 2.0 m/s or 2.5 m/s, for the first and second day, respectively. During warm-up, participants breathed through the expired-gas analysis system to allow acclimatization. For each running velocity, the different incline conditions were randomized to prevent order effects.

We measured the rates of oxygen consumption ($\dot{V}O_2$) and carbon dioxide production ($\dot{V}CO_2$) using an open-circuit expired-gas analysis system (True One 2400, Parvo Medics, Salt Lake City, UT, USA). We calibrated the gas analyzers before each test using reference gases. The flow-rate transducer was calibrated using a 3 liter syringe (Rudolph Inc., Kansas City, MO, USA). Each trial lasted 7 min based on pilot data showing that steady state was reached in less than 5 min during the different trials. We averaged $\dot{V}O_2$, $\dot{V}CO_2$ and respiratory exchange ratios (RER) for the last 2 min of each trial. Rest periods of at least 4 min occurred between the trials. During the rest periods, the treadmill was adjusted to the incline and velocity of the following trial.

## Calculations

To fit a generic curve to the wasted impulse data (Eq. (6)), we first normalized the impulse data to body mass and divided the values by running velocity, similar to the cost of transport concept (see below). Mechanical vertical COM power (in Watts) was calculated using belt velocity and incline (similar to Eq. (2)):

$$\textit{Mechanical vertical COM power} = m \cdot g \cdot sin(\theta) \cdot v \qquad (7)$$

where $\theta$ is the incline in degrees and $v$ is velocity in m/s. Metabolic rates (in W/kg) were calculated from respiratory measurements using the Brockway equation (*Brockway, 1987*). Net metabolic power was calculated as running metabolic rate minus the standing metabolic rate. We calculated the traditional values of delta efficiency of producing mechanical power to lift the COM vertically as the ratio of mechanical vertical COM power to the difference in metabolic rate between level running and running on incline at the same velocity (*Gaesser & Brooks, 1975*).

Net metabolic Cost of Transport (CoT) is the net metabolic cost per unit distance traveled parallel to the running surface. It is calculated by dividing the net metabolic rate by the running velocity and is expressed in J/(kg m). Cost of Transport values allowed us to develop a generalized equation, independent of running velocity. Based on the general concepts underlying our uphill running model, we generated a custom equation and fitted this to the data to calculate the parameters resulting in the best fit (see below). The format of the equation is:

$$Net\ CoT\ (J/(kg \cdot m)) = A + B \cdot e^{-\lambda \cdot sin(\theta)} + \frac{g}{Eff_{vCOM}} \cdot sin(\theta) \tag{8}$$

In this equation, the CoT of parallel running is represented by $A + B \cdot e^{-\lambda \cdot sin(\theta)}$. We postulated that the cost of parallel running decreases exponentially with incline. We expected that at steep inclines, where $I\_GRF_{wasted}$ equals zero, the cost of braking and propelling would be reduced to zero and that the cost of parallel running would consist of only the costs of perpendicular bouncing, leg swing and arm swing. In terms of our model, the first term $A$ represents the CoT related to perpendicular bouncing, leg swing and arm swing. Parameter $B$ represents the CoT for braking and propelling during level running. For inclined running, the CoT for braking and propelling parallel to the running surface decreases exponentially with the sine of the incline angle $\theta$: $CoT_{braking/propelling} = B \cdot e^{-\lambda \cdot sin(\theta)}$. The decay constant $\lambda$ determines how steeply the $CoT_{braking/propelling}$ decreases with $sin(\theta)$. Logically, the $CoT_{braking/propelling}$ decreases proportionally to the wasted GRF impulse per step $I\_GRF_{wasted}$, i.e., that $\lambda$ in Eq. (8) is equal to $\gamma$ in Eq. (6).

The CoT of producing mechanical power to lift the COM vertically is represented by the third term in Eq. (8). To relate the mechanical vertical COM power (Eq. (7)) to the metabolic CoT, it should be divided by body mass, velocity and the efficiency of producing mechanical power to lift the COM vertically, resulting in $\frac{g}{Eff_{vCOM}} \cdot sin(\theta)$.

### Statistical analyses

We present all results in the text as mean values $\pm$ SD. We used a traditional level of significance ($\alpha = 0.05$) for all statistical tests. To test for significant differences between the three tested running velocities and between different angles, we applied two-way analyses of variance (ANOVAs) on the impulse, step frequency and contact time data. We applied the non-linear least squares method to fit non-linear curves on the data and the linear least squares method to fit lines. We utilized $r^2$ to evaluate goodness of fit.

## RESULTS

### Ground reaction forces

For running at a velocity of 2.0 m/s, the braking GRF impulse per step, parallel to the running surface, normalized to body mass and divided by the running velocity, decreased significantly from −0.128 for level running to −0.003 for running up a 9° incline. For 2.5 m/s and 3.0 m/s similar decreases were observed (Table 1). For two participants, we

**Table 1** Braking, total propelling and wasted impulses, step frequencies and contact times (mean ± SD) for the different test conditions.

| | | Level | 9° |
|---|---|---|---|
| Braking Impulse ($10^{-3}$) | 2.0 m/s | $-63.9 \pm 11.9$ | $-1.7 \pm 1.7$ |
| | 2.5 m/s | $-58.3 \pm 10.8$ | $-3.4 \pm 2.6$ |
| | 3.0 m/s | $-55.1 \pm 8.8$ | $-3.5 \pm 2.2$ |
| Total Propelling Impulse ($10^{-3}$) | 2.0 m/s | $64.3 \pm 12.2$ | $0.9 \pm 1.3$ |
| | 2.5 m/s | $58.5 \pm 10.7$ | $2.0 \pm 2.5$ |
| | 3.0 m/s | $55.3 \pm 8.7$ | $2.3 \pm 2.1$ |
| $I\_GRF_{wasted}$ ($10^{-3}$) | 2.0 m/s | $128.2 \pm 24.0$ | $2.6 \pm 2.9$ |
| | 2.5 m/s | $116.8 \pm 21.5$ | $5.4 \pm 5.0$ |
| | 3.0 m/s | $110.3 \pm 17.4$ | $5.7 \pm 4.2$ |
| Step freqency (steps/second) | 2.0 m/s | $2.68 \pm 0.15$ | $2.72 \pm 0.18$ |
| | 2.5 m/s | $2.78 \pm 0.20$ | $2.84 \pm 0.16$ |
| | 3.0 m/s | $2.84 \pm 0.19$ | $2.94 \pm 0.16$ |
| Contact times (s) | 2.0 m/s | $0.31 \pm 0.03$ | $0.32 \pm 0.03$ |
| | 2.5 m/s | $0.28 \pm 0.03$ | $0.28 \pm 0.03$ |
| | 3.0 m/s | $0.25 \pm 0.02$ | $0.25 \pm 0.02$ |

**Table 2** Measured rates of oxygen consumption ($\dot{V}O_2$) and metabolic rates (mean ± SD) for the different test conditions.

| Day | Velocity (m/s) | Angle (degrees) | Grade (%) | $\dot{V}O_2$ (ml/(kg min)) | Metabolic rate (W/kg) |
|---|---|---|---|---|---|
| 1 | | Standing | – | – | $4.3 \pm 0.5$ | $1.5 \pm 0.1$ |
| | 2.0 | 0 | 0 | $24.5 \pm 1.5$ | $8.3 \pm 0.4$ |
| | | 1 | 1.7 | $26.1 \pm 1.2$ | $8.9 \pm 0.4$ |
| | | 2 | 3.5 | $29.1 \pm 1.3$ | $9.9 \pm 0.4$ |
| | | 3 | 5.2 | $31.6 \pm 1.8$ | $10.8 \pm 0.5$ |
| | | 4 | 7.0 | $34.3 \pm 1.1$ | $11.7 \pm 0.4$ |
| | | 6 | 10.5 | $40.5 \pm 1.8$ | $13.9 \pm 0.6$ |
| | | 8 | 14.1 | $47.1 \pm 2.1$ | $16.3 \pm 0.7$ |
| 2 | | Standing | – | – | $4.7 \pm 0.4$ | $1.6 \pm 0.1$ |
| | 2.5 | 0 | 0 | $29.0 \pm 1.3$ | $9.8 \pm 0.4$ |
| | | 1 | 1.7 | $31.6 \pm 1.4$ | $10.7 \pm 0.4$ |
| | | 2 | 3.5 | $35.4 \pm 1.4$ | $12.0 \pm 0.4$ |
| | | 3 | 5.2 | $38.3 \pm 1.2$ | $13.1 \pm 0.4$ |
| | | 4 | 7.0 | $42.2 \pm 1.0$ | $14.4 \pm 0.3$ |
| | 3.0 | 0 | 0 | $35.3 \pm 1.6$ | $11.9 \pm 0.5$ |
| | | 1 | 1.7 | $38.9 \pm 2.0$ | $13.2 \pm 0.6$ |
| | | 2 | 3.5 | $43.1 \pm 1.8$ | $14.7 \pm 0.6$ |
| | | 3 | 5.2 | $47.1 \pm 1.7$ | $16.1 \pm 0.5$ |
| | | 4 | 7.0 | $51.6 \pm 2.2$ | $17.8 \pm 0.7$ |

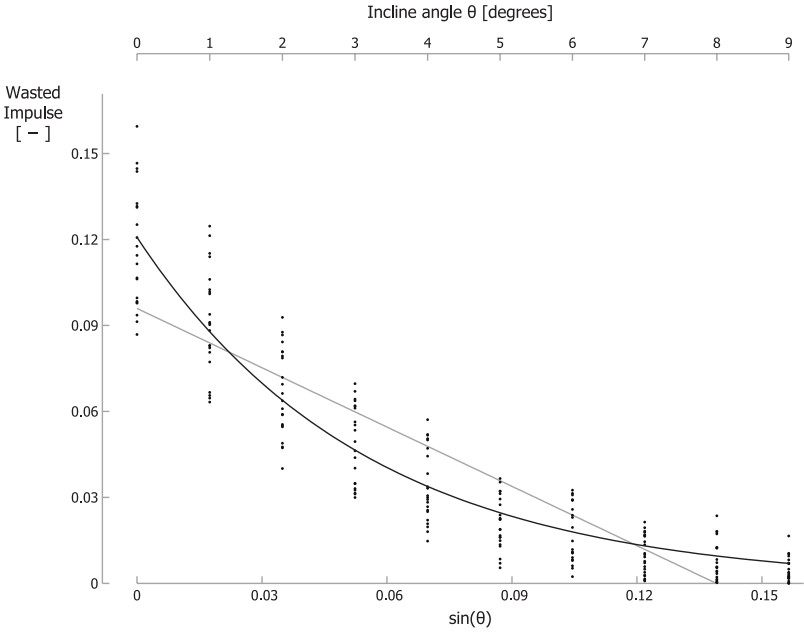

**Figure 3** **The wasted GRF impulse per step decreases for increasing inclines.** Wasted GRF impulse for different inclines and velocities. Each small dot represents a single participant's wasted GRF impulse for a specific trial. The black line is the best-fit curve to the data according Eq. (6); the grey line is the best-fit linear equation. Note that the secondary horizontal axis at the top is the incline angle $\theta$, which is not linear, so the tick marks are not evenly spaced.

could not analyze the kinetic data at 2 m/s since they "ran" without a clear aerial phase at this velocity. As such, their stance phases partially overlapped, invalidating the assessment of the braking and propelling impulses during each stance phase. In addition, for the same reason, we excluded 4 separate trials for other participants. Similar to the braking impulse, the total propelling impulse per step, parallel to the running surface, and the wasted GRF impulse per step ($I\_GRF_{wasted}$), also decreased with incline (Table 1; Fig. 3; individual trial data is contained in Supplemental Information 1). Recall that total propelling impulses were calculated as the propelling GRF impulse parallel to the surface minus the component of the gravitational impulse parallel to the running surface. Summation of the absolute values of the braking impulse and of the total propelling impulse per step gave $I\_GRF_{wasted}$. In line with our hypothesis, $I\_GRF_{wasted}$ values decreased exponentially with incline. Curve fitting of Eq. (6) to the GRF data resulted in best-fit parameter values of $I\_GRF_{wasted\ level} = 0.1208$ and $\gamma = 18.24$, with $r^2 = 0.89$ (Fig. 3). Fitting a linear equation to the data resulted in a lower correlation between the data and the fit ($r^2 = 0.79$) and implied negative $I\_GRF_{wasted}$ values for inclines steeper than about 8 degrees.

The step frequency increased significantly with incline and with running velocities (Table 1). In contrast, contact times were similar between inclines ($p = 0.7$) and decreased significantly with velocity (Table 1).

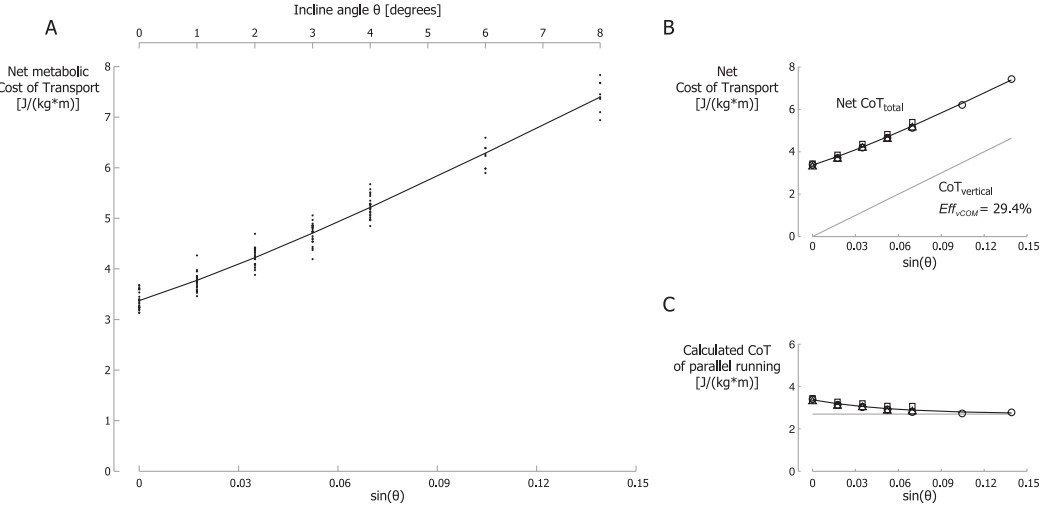

**Figure 4 Net metabolic Cost of Transport.** (A) Net metabolic Cost of Transport (CoT) for different inclines and velocities. CoT is the net metabolic energy consumed per meter traveled parallel to the running surface. Each small dot represents a single participant's CoT for a specific trial. The black line is the best-fit curve of the net CoT according equation 8. Note that the secondary horizontal axis at the top is the incline angle $\theta$, which is not linear, so the tick marks are not evenly spaced. (B) Net metabolic CoT$_{total}$ and metabolic CoT$_{vertical}$. (C) Metabolic CoT of parallel running. The grey line represents the constant CoT components of parallel running (perpendicular bouncing, leg and arm swing and lateral balance); the remainder, the CoT of braking and propelling, approaches zero at steeper inclines. Symbols represent mean values: $\circ$, 2.0 m/s; $\triangle$, 2.5 m/s; $\square$, 3.0 m/s.

## Metabolic measurements

In Table 2 we present the mean rates of oxygen consumption ($\dot{V}O_2$) and metabolic energy consumption (individual trial data is contained in Supplemental Information 2). For all participants, RER ($0.86 \pm 0.05$; range 0.74–0.96) was less than 1.0 for all trials, indicating that the metabolic energy was derived primarily from oxidative sources. The mean metabolic rate for standing was $1.53 \pm 0.08$ W/kg.

## Decreasing cost of parallel running

Net metabolic Cost of Transport (CoT) data for all participants are shown in Fig. 4A for different inclines and velocities. The net CoT data are plotted versus the sine of the incline angle $\theta$ on the primary horizontal axis (at the bottom) because the vertical power is proportional to the sine of the incline angle. The net CoT is the net metabolic cost expressed per unit distance traveled parallel to the running surface.

We set $\lambda$ in our model (Eq. (8)) to be equal to $\gamma$ (from Eq. (6)) and calculated the best fit to the metabolic data. The parameter of the best fit with $\lambda = \gamma = 18.24$ were $A = 2.70$, $B = 0.674$ and $Eff_{vCOM} = 29.4\%$ with $r^2 = 0.97$. The best-fit curve is shown in Fig. 4A. In Fig. 4B this best-fit curve is labeled Net CoT$_{total}$ as it includes all terms of Eq. (8), i.e., the CoT of parallel running and the CoT of producing mechanical power to lift the COM vertically. The net CoT data are shown as mean values for each running

velocity in this figure (Fig. 4B). The metabolic CoT of producing mechanical power to lift the COM vertically is also shown (labeled CoT$_{vertical}$). This CoT was calculated based on $Eff_{vCOM} = 29.4\%$. In our model, the CoT of parallel running is represented by $A + B \cdot e^{-\lambda \cdot sin(\theta)}$, and this cost is shown in Fig. 4C for $A = 2.70$, $B = 0.674$ and $\lambda = 18.24$. Metabolic data points were calculated by subtracting the calculated metabolic CoT of producing mechanical power to lift the COM vertically from the net CoT. This resulted in the following equation:

$$Net\ CoT(J/(kg \cdot m)) = 2.70 + 0.674 \cdot e^{-18.24 \cdot sin(\theta)} + \frac{g}{0.294} \cdot sin(\theta) \tag{9}$$

Note that the best-fit regression for the net CoT versus the sine of the incline angle $\theta$, using Eq. (8), is fairly insensitive to changes in the parameters. For instance, curve fitting of Eq. (8) with $\lambda$ as a free parameter resulted in $A = 1.16$, $B = 2.20$, $\lambda = 7.60$ and $Eff_{vCOM} = 24.9\%$ and produced a similar goodness of fit: $r^2 = 0.97$.

## DISCUSSION

In this study, we quantified the ground reaction forces and metabolic cost of uphill human running and introduced a new model to interpret our results. This is the first model for uphill running that incorporates the cost of generating force concept. We have found that the metabolic rate during uphill running can be predicted by a model which posits that (1) the metabolic cost of perpendicular bouncing remains the same as during level running, (2) the metabolic cost of running parallel to the running surface decreases with incline, (3) the delta efficiency of producing mechanical power to lift the COM vertically ($Eff_{vCOM}$) is constant, independent of incline and running velocity, and (4) the costs of leg and arm swing do not change with incline.

### Ground reaction forces

The GRF data confirmed that the wasted braking and propulsive impulses per step decrease exponentially with incline supporting our contention that the metabolic cost of parallel running decreases with incline. Based on this, we generated a general model for the metabolic cost of uphill running.

### Metabolic cost of uphill running

In line with earlier observations of net mechanical efficiency values approaching the efficiency of concentric contracting muscles on steeper inclines (*Margaria, 1968*; *Margaria et al., 1963*), our model assumes that the efficiency of producing mechanical power to lift the COM vertically ($Eff_{vCOM}$) is constant, independent of incline and running velocity, and physiologically realistic. Our method offers an alternative to the model by Minetti and co-workers (*Minetti, Ardigò & Saibene, 1994*; *Minetti et al., 2002*) which assumed that the metabolic cost can be predicted completely based on measures of mechanical work. In contrast, our model combines the cost of generating force to support the runner's body weight and the cost of performing mechanical work to lift the COM. In our approach, the different terms in the model each represent different elements of the CoT

of uphill running. Unfortunately, the CoT of each of these elements cannot be measured independently. Therefore, we constructed a biomechanical realistic model and applied a fitting procedure to calculate the parameters needed.

## Metabolic cost of parallel running

According to Eq. (9), for level running ($\theta = 0; sin(\theta) = 0$), about 80% of the net metabolic CoT would be attributed to weight support (perpendicular bouncing), leg swing and arm swing, while 20% would be attributed to braking and propelling the COM. These number relate well with earlier studies on the cost of supporting body weight (at most 74% of the net cost of running; *Teunissen, Grabowski & Kram, 2007*) and of leg swing (only ∼10% of the net cost of running; *Moed & Kram, 2005*), which sum up to ∼84% of the net metabolic cost attributable to weight support and leg swing.

In our model, the CoT related to perpendicular bouncing, leg swing and arm swing is independent of incline. However, step frequency increased slightly with incline, which could result in higher values for "internal work" (*Minetti, 1998*) or joint mechanical power (*Swanson & Caldwell, 2000*). We estimated mechanical internal work values (in J/(kg m)) based on step frequency, duty factor and velocity as per the *Minetti* equation (*1998*), using different values for factor $q$ for level and uphill running (*Nardello, Ardigò & Minetti, 2011*). These estimates of mechanical internal work increased both with incline and running velocity. Although the *Minetti* equation (*1998*) suggests that internal power would increase by 37% between 2.0 and 3.0 m/s, we did not observe any change in the overall metabolic CoT. Similar increases in the internal mechanical power were estimated between level and uphill running (32–33%, for our range of velocities). It is unclear how these mechanical internal work estimates relate to the metabolic CoT because of overestimations of internal work related to the ballistic pendulum-like part of the swing phase of the limbs (*Alexander, 1989*). Furthermore, *Nardello, Ardigò & Minetti (2011)* reevaluated the 1998 Minetti equation for humans of both sexes, for different age groups, running at different velocities and inclines and they observed no increase in measured internal work as function of incline for velocities below 2.78 m/s. Additionally, evidence from our laboratory suggests that the metabolic cost of leg swing in human running is relatively small, ∼10%–20% of net metabolic cost of running (*Modica & Kram, 2005*; *Moed & Kram, 2005*). Finally, guinea fowl blood flow data suggest that the majority of the increased energy expenditure in uphill running is used by stance phase muscles (*Rubenson et al., 2006*). Thus, for simplicity in our model, we assumed that the cost of leg swing is independent of incline.

It is difficult to estimate the metabolic cost of arm swing. Experiments that restrict arm swing increase the cost of running by at least 3% (*Arellano & Kram, in press*) suggesting that arm swing produces a net energy savings rather than a net cost. In any case, it seems unlikely that the metabolic cost or savings due to arm swing at a certain running speed would change greatly during uphill running. Thus, we subsume the cost of arm swing into the cost of perpendicular bouncing and assume that it does not change.

## Metabolic cost of producing mechanical power to lift the COM vertically

The CoT of producing mechanical power to lift the COM vertically increases linearly with $sin(\theta)$, proportional to the mechanical vertical COM power. This is a direct consequence of our assumption that the efficiency of producing mechanical power to lift the COM vertically ($Eff_{vCOM}$) is constant, independent of incline and running velocity. The efficiency of producing mechanical power to lift the COM vertically ($Eff_{vCOM}$), according the best fit of our model was 29.4%. This value is in the same range as earlier reported values of similar measures of whole body efficiency in cycling. *Gaesser & Brooks (1975)* defined work efficiency as work accomplished divided by the energy expended above that in cycling without a load. They found values ranging from 25.4% to 30.3% for increasing cadence and power output. *Bijker, De Groot & Hollander (2001)* reported a mean delta efficiency (delta work accomplished over delta energy expended) of 25.8% in ergometer cycling. In contrast, Margaria's net mechanical efficiency (vertical mechanical power/net metabolic rate) values were rather low ($\sim$9%–16%; *Minetti et al., 2002*) for running up inclines typical of recreational running. Alternatively, the traditional vertical efficiency (vertical mechanical power/difference in metabolic rate between locomotion on an incline and level locomotion at the same velocity) and similarly calculated measures result in high values ($\sim$36%–46%; *Asmussen & Bonde-Petersen, 1974*; *Bijker, De Groot & Hollander, 2001*; *Cooke et al., 1991*; *Lloyd & Zacks, 1972*; *Pugh, 1971*).

## Limitations and future directions

Our study has several limitations worthy of mention. As discussed earlier, we performed the two parts of the study (GRF and metabolic data collection) with two different groups of participants. We acknowledge this as a limitation of the study, however, because our model parameters were calculated using regression equations for group data we consider this not to be a serious concern. Further, we are not attempting to make subject specific conclusions, rather we are seeking general principles. Overall, we were limited by the aerobic capacity of the participants. We tried to include a broad range of velocities and inclines, but we were restricted by our aim to consider only conditions that could be run at truly submaximal intensities by all our participants. Although we did not quantify the elastic energy storage and reutilization, we accounted for this by introducing the cost of perpendicular bouncing, which we assumed to be independent of incline and proportional to velocity.

The, overall, promising agreement between the experimental data and the equations based on the assumptions underlying our approach, call for further validation of this approach in future studies. Addressing any effects on cost of potential changes in internal work (CoT of leg swing), mechanical joint work and joint posture could refine the accuracy of and increase the confidence in our approach. It would be interesting to study the energetics of uphill walking with the same approach as we have done here for running. More insights into the energetics of downhill running may be gained with our approach. Of course, our concept of decreased parallel braking impulses would need to be reversed.

## CONCLUSIONS

Overall, we postulate that the metabolic rate during uphill running is not simply equal to the sum of the cost of level running and the cost of performing work to lift the body mass against gravity. Rather, our new approach suggests that the metabolic cost of running at a certain velocity, parallel to the running surface, decreases with incline, and that the efficiency of producing mechanical power to lift the COM vertically is constant, independent of incline and running velocity. With this approach, we have been able to model the observed metabolic rates during uphill running at different velocities and inclines.

## ACKNOWLEDGEMENTS

We thank Dr. Maarten Bobbert and members of the Locomotion Lab of the University of Colorado Boulder for insightful comments and suggestions.

### Funding

This study was financially supported by the GJ van Ingen Schenau Promising Young Scientists Award that was awarded to W Hoogkamer by the Faculty of Human Movement Sciences at the VU University of Amsterdam. The funders had no role in study design, data collection and analysis, decision to publish, or preparation of the manuscript.

### Grant Disclosures

The following grant information was disclosed by the authors:
GJ van Ingen Schenau Promising Young Scientists Award.

### Competing Interests

The authors declare there are no competing interests.

### Author Contributions

- Wouter Hoogkamer conceived and designed the experiments, performed the experiments, analyzed the data, contributed reagents/materials/analysis tools, wrote the paper, prepared figures and/or tables, reviewed drafts of the paper.
- Paolo Taboga and Rodger Kram conceived and designed the experiments, performed the experiments, analyzed the data, contributed reagents/materials/analysis tools, wrote the paper, reviewed drafts of the paper.

### Human Ethics

The following information was supplied relating to ethical approvals (i.e., approving body and any reference numbers):
0606.29 University of Colorado Boulder IRB.

**Peer**J ______________________________________________

## Supplemental Information

Supplemental information for this article can be found online at http://dx.doi.org/10.7717/peerj.482.

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
