# Peer review of "Applying the cost of generating force hypothesis to uphill running"

_PeerJ, doi:10.7717/peerj.482_

## Round 0.1 · original submission · Major Revisions

It is clear that the reviewers appreciate the design and reporting of your study. However, there are certain clarifications, reporting modifications, and perhaps some methodological adjustments that need to be carried out.

·

Basic reporting

The Introduction eventually (see next comment) did a good job of motivating the need for a new model of uphill running energetics, but I didn’t think it motivated the importance/interest of studying uphill running energetics in the first place. I think it would be good to add a brief explanation, probably early on, on why the topic is of interest / importance.

The Introduction was really long (1600 words by my count). I found it a chore to get through and figure out what the actual problems/questions were. I really don’t think you need such a long and detailed literature review with so many specifics on the methods and results of individual papers. I suggest shortening it up by removing some of the specifics on the previous studies and emphasizing (i) what their limitations were and (ii) how this new model addresses them.

Somewhere in the Introduction, I think it would be helpful to briefly summarize the “Cost to generate force” hypothesis for running (Kram & Taylor, 1990). It is mentioned and seems to be a major component of the new model (Line 85) but how it relates specifically to the new model and addresses limitations of the previous models wasn’t clear to me.

The authors did a good job in most places of being specific about which particular work-related quantity among the many used in movement science they were using or referencing, but throughout the paper, please check the use of generic terms like mechanical work/power/etc. and ensure that the specific work-related quantity in question (e.g. external work, joint work, etc.) is noted with a specific and consistent use of the terminology. Relatedly, I think the paper would benefit from an explanation of the choice of external work vs. some other work quantity in the efficiency calculations. For example, none of the typical mechanical work quantities from motion analysis (external/internal/joint work) relate particularly well to the work done by muscles (Gregor et al., 1991; Prilutsky et al., 1996; Neptune & van den Bogert, 1998; Sasaki et al., 2009). If the implicit or explicit purpose of the analysis is to quantify muscle work, then I think this should be noted and discussed as a limitation.

Line 44: It wasn’t intuitively obvious to me why the existing models are invalid for shallower inclines. Is it because the inclines are too shallow to assume that most of the work is concentric? If so, or whatever the reason may be, I think it would help for clarity. I also suggest avoiding direct quotations (from the Minetti paper) but that’s more of a stylistic suggestion. Also be careful with equating reliability (Line 42) with validity (45); they are not the same, as I’m sure the authors are aware.

Line 57: “Work efficiency is thereby similar to vertical efficiency…” From the descritions presented, it sounds to me like they’re not just similar, they’re identical: both are the external work rate over the increase in metabolic rate relative to level running. Did I miss a distinction between the two?

Line 90: Re: perpendicular bouncing: I didn't follow this statement and I think it’s an important one in the context of this work. It sounds like the vertical direction (supporting the body weight) in level running is being equated with the surface perpendicular direction in uphill running; is this correct? Regardless of the surface grade, the vertical direction along which the body weight is supported is always the global vertical direction, so how can it be equated with another direction?

Line 109: “During steeper inclined running, because most of the propulsive parallel GRF impulse is required to overcome the component of the gravitational braking impulse parallel to the surface.” I think a word might be missing somewhere from this sentence?

Experimental design

Line 146: 20 Hz seems really low for GRF in running. Does this choice make a difference in the results?

Line 155: It wasn’t clear to me how this sum of absolute impulses represents a “wasted” impulse quantity. Could the authors expand on the mechanical rationale for this calculation? In Line 160 with the wasted impulse in level running, I really got lost. Here the gravitational component is zero, so the “wasted” impulse is the sum of the absolute braking and propelling impulses? I’m not following what this quantity represents physically or how it is “wasted” exactly. Also why is an exponential model assumed (e.g. why not quadratic, etc.)?

Line 213: Recent evidence has suggested the CoT is not independent of running speed (Steudel-Numbers & Wall-Scheffler, 2009), although it is much less sensitive to speed than walking.

Line 241: With such a small sample, I’m not sure there’s any point in testing for sex differences.

Validity of the findings

The journal asks that these data are provided or made available, I didn't see any mention of this availability in the paper though.

Line 304: In line with my previous comment, it wasn’t clear to me how the cost of generating force concept is included here.

Line 305: “…can be accurately predicted…” You only tested this one model though. How can you know how “accurate” it is? Accurate in comparison to what? You performed a curve-fitting (least-squares) exercise so of course the model has to fit the data well if you give it enough free parameters, that will always be the case, but that does not validate the model in and of itself. Validation I think would require testing the model’s predictions against something you already know the answer to, without fitting the model parameters specifically to match that answer as well as it can.

Lines 311-320: This is duplicated from the Results section, no need to present it again here.

Line 328: This may be true (hard to say) but you don’t have data to back it up one way or the other. Rather than claiming it is true, I think it would be more appropriate to state that the use of two different subject groups was a limitation, then rationalize why you don’t think it was a critical one.

Along the same lines, why were two different groups used anyway? Why not just have the subjects from the second experiment run on the force treadmill so you don't have to combine GRF and CoT data from two different groups with different gender breakdowns, masses, fitness levels, etc.?

Lines 345-351: These don’t seem like discussion points to me. Shouldn’t this explanation of what the model’s terms represent be presented in the Methods, when the model is presented?

Line 379: I’m not familiar with this reference (Tseh et al., 2008), but another study (Arellano & Kram, 2011) found an 8% increase in metabolic power when arm swing was suppressed. This doesn’t necessarily mean arm swing accounts for 8% of the metabolic cost of running, but it’s hardly an insubstantial amount. Again, I suggest noting this is a limitation of the model rather than speculating that it’s not important.

Lines 391-399: Why are these values from other models deemed physiologically unrealistic? Is it because they are different from the usual ~ 25% value found for concentric muscle actions?

Line 408: I don't think this is necessarily true; see previous comment on work-related quantities. External work is not muscle work and vice-versa, they will only ever be equal by coincidence, although they may scale with one another in certain situations.

Line 439: I don’t think you have actually found this as it wasn’t tested here. Only one model was tested. As I suggested above, this model has to fit the data well. There is no assurance that another model also wouldn’t fit it well. If you want to show that this model works better than another, you should test that other model on these data too.

Additional comments

References from the reviewer comments:
Arellano CJ and Kram R (2011). The effects of step width and arm swing on energetic cost and lateral balance during running. Journal of Biomechanics 44, 1291-1295.

Gregor RJ, Komi PV, Browning RC, and Järvinen M (1991). A comparison of triceps surae and residual muscle moments at the ankle during cycling. Journal of Biomechanics 24, 287-297.

Kram R and Taylor CR (1990). Energetics of running: a new perspective. Nature 346, 265-267.

Neptune RR and van den Bogert AJ (1998). Standard mechanical energy analyses do not correlate with muscle work in cycling. Journal of Biomechanics 31, 239-245.

Prilutsky BI, Petrova LN, and Raitsin LM (1996). Comparison of mechanical energy expenditure of joint moments and muscle forces during human locomotion. Journal of Biomechanics 29, 405-415.

Sasaki K, Neptune RR, and Kautz SA (2009). The relationships between muscle, external, internal and joint mechanical work during normal walking. Journal of Experimental Biology 212, 738-744.

·

Basic reporting

In general I believe the basic reporting of this manuscript to be good. All the important information is included, the study is well introduced with an appropriate synthesis of relevant literature that highlights the rationale for this research. In terms of reporting, I only have a couple of comments aimed at improving the clarity/readability of the manuscript:

1. I believe that the clarity of the manuscript would be significantly helped by a section or schematic figure that breaks down the components of the model. While all the model components are defined at some point in the methods, I found it took me quite some time to pull them all together and decipher what mechanical quantities were included in each component. For example, it was not clear initially that the 'cost of parallel running' included the 'cost of braking and propelling', the 'cost of perpendicular bouncing' and the 'cost of leg and arm swing'. The terminology makes sense but, I think it would benefit from being clearly spelled out in a figure (or at least have all the terms clearly defined in one place - e.g. a glossary of terms / appendix).

2. On page 7, the description of the calculation of wasted impulse might benefit from being more in equation form rather than a paragraph of text. The text was not easy to follow.

3. A figure showing the fit (and goodness of fit) of an exponential function to the GRF (impulse) data that provided the value for γ. This value is important in the model and the reader should have some means of evaluating it's appropriateness.

Experimental design

With regards to experimental procedures i believe the authors have treated data appropriately and that the model is interesting and appropriate, with correct equations. As the authors point out in the discussion, the number of experimental conditions (inclines) is somewhat limiting and although, I do not consider this a fatal flaw of the study i do think it requires some more attention with regard to it's implications for the model (see comment below).

My main issue is that the authors seem to have arbitrarily arrived at the decision to use an exponential fit to describe the decline in cost associated with braking and propulsion. While I agree that this cost should become less with increasing incline, I see no obvious rationale for why this should be an exponential function (I also do not believe the authors present any rationale). If you had more experimental data points to fit your curve to, you might be able to provide a convincing argument based on goodness of fit. however, with only two experimental conditions I fail to see how an exponential function would provide a better fit than a linear function. What was the rationale for using an exponential function? Was it simply that it produced a more believable (and constant) efficiency of positive work term? I think the rationale should be provided in the manuscript and the consequences of using other types of function discussed. For example, how sensitive or robust is the model to this choice of function?

Another issue is the assumption that the cost of supporting body weight (or perpendicular bouncing) is assumed to be constant across conditions. This seems a reasonable assumption for changing incline where contact times and force peaks are relatively constant and the authors note this. However, contact time decreased significantly with velocity and this seems to be somewhat ignored. As I am sure the authors would agree, contact time dictates the time available to produce force to support body weight. Thus, as contact time becomes less, the rate of force production must increase (as the average force to support body weight remains the same). This increased rate of force production should (according to cost of force hypotheses) increase the cost of supporting body weight (perpendicular bouncing). It seems to me, that this might require the model to have a cost of perpendicular bouncing that varies with velocity. Why was this not included in the model?

Validity of the findings

With regard to the validity of the findings, I am happy that the the authors' conclusions are appropriate based on their data. However, I would like to see them respond to my above comments to review the potential impact of those issues on the output of the model.

Additional comments

The authors should be congratulated on producing an interesting and novel approach to understanding the mechanisms behind observed trends in the metabolic demands of uphill running. In general I enjoyed reading the manuscript and think that it represents a significant advance on previous attempts to predict the efficiency of work in this task. In the other sections of this review I have outlined some queries I have regarding the model itself which I would like the authors to respond to before I would recommend publication. I have also made some suggestions for improving the manuscript's clarity.

·

Basic reporting

The manuscript is well written and structured. Below are some minor comments:

p. 2 line 32-34. I’m not sure what is meant by “even though it is inevitable”.

p. 6 line 127. Did you mean metabolic costs were measured at 8 degrees at velocities up to 3 m/s (seems aerobically tough!). Or that 2m/s was tested at 8 deg?

p.7 lines 156 – 160. Cumbersome sentence.

p.8 line 168. Were subjects for the force experiments as fit as those for the metabolic measurements?

p. 10 line 215. How was data fitted? Linear least-squares? Or some other technique?

p. 14, lines 312 – 319. This seems to fit better in methods/results.

p. 16, lines 347 – 351. Seems a bit repetitive.

p. 16, mechanical joint work in limb swing has also been found to increase with incline in humans (Swanson and Caldwell, 2000, MSSE).

Experimental design

Overall the experimental design and methodology are appropriate and address the hypotheses well. The following questions and comments are offered as constructive suggestions to the authors for improving the overall quality of the manuscript.

1. One of the hypotheses is that the wasted GRF impulse per step decreases exponentially with steeper inclines. As the authors note, the accuracy of fitting an exponential equation to the two tested inclines (0 and 9 deg.) is questionable. I’m wondering if the authors thought of fitting data from their previous studies (Gottschall and Kram, 2005, Snyder et al., 2012) that include 0, 3, 6 and 9 deg. Considering that the study is already combining mechanical and metabolic data sets from different subjects, I think this approach would be reasonable. In this case the study would be partly a meta-analysis of sorts, but I don’t see anything wrong with that.

2. The authors claim on page 5 that because the peak forces and contact times do not change with incline that the cost of force during vertical bouncing should remain constant. But what about the joint’s effective mechanical advantage (joint posture)? Don’t these also affect cost of force, and are they also constant with incline? Perhaps mention this here or in limitations section on p19.

3. The model assumes a constant cost of limb swing in uphill vs level running. Albeit in a bird, the blood flow data from Rubenson et al. 2006 (JEB) support the theory that the increase in cost when running uphill is mainly the result of stance mechanics, although they found an 11% increase in swing cost. A similar increase might also happen in humans given that mechanical work in swing is known to go up slightly.

I think the model is fine as is, but wonder if adding a cost of swing term in the equation can be beneficial. If it is kept constant (e.g. 20% based on Modica and Kram) it can provide a cost specific to vertical bouncing. It could also allow a means to test the sensitivity of the constant swing cost assumption.

4. Figure 2: could the cost of bouncing and cost of breaking-propelling also be disintegrated in panel C?

Validity of the findings

The data presented in the study are sound and add an interesting perspective on incline running energetics. My main comments are around the conceptual framework combining cost of force and work in the context of uphill running:

1. The study aims to generate a model of uphill running energetics that “combines the cost of generating force and the cost of performing mechanical work”. Does ‘cost of generating force’ specifically refer to force generated to support the body during ‘perpendicular bouncing’ that occurs in the absence of muscle work? (if not, then the force is a component of work and the distinction of ‘force’ and ‘work’ would, in my opinion, lose its value). If I have interpreted this correct, it may be worth stating that the model assumes the muscular mechanisms of the bouncing remain constant across incline- i.e. force production without work- with a separate additional component of work to raise the center of mass uphill. This is plausible, although I am not familiar with much in vivo data that shows this. I don’t know of studies that show muscles functioning isometrically uphill, but you may want to cite Lichtwark and Wilson (2006, JEB) that showed that the medial gastroc. functions pseudo-isometrically in early stance followed by shortening during plantarflexion in both level and uphill running. Also Gabaldon and Roberts data (2004) hint at this possibility in turkey ankle extensors. But how much of the muscle fiber work in these studies is specifically to do work to raise the COM vertically or due to deceleration and re-acceleration of the body during the step is unclear.

2. This study assumed a proportion of the metabolic cost is for functions other than vertical COM mechanical work and estimated the efficiency of the remaining cost of vertical COM work using a data fitting procedure. This resulted in a good agreement with the hypothesis, and that the fit converged on a realistic muscle efficiency is intriguing. But this fitting approach provides limited mechanistic support for the different components of uphill running costs. The authors do a pretty good job not over selling the data fitting results, but in a few spots some caution should be considered when claiming that best-fit-parameters explain determinants of uphill running (e.g. concluding statement of abstract).

Of course, the results also do not preclude other mechanistic explanations for the energy cost of uphill running. While I do not disagree that cost of force may play a role, uphill costs may follow other paradigms. For example, cost may be set by the total amount of mechanical work done by the muscle fibers, including mechanical work done during deceleration/acceleration of the COM in each step plus that required to raise the COM uphill. Elastic (‘vertical bouncing’) mechanisms (e.g. tendon stretch-recoil) may still exist, but they do not depend necessarily on isometric muscle force production. In fact, there is fairly strong data indicating that the gastrocnemius and soleus in humans (muscles typically thought of in terms of isometric strut function) mostly shorten and do positive fiber work during stance in running, even level running (although the overall mechanical work at the ankle is still shared with stretch-recoil of the tendon) (Lichtwark et al, 2005 J. Biomech.; Rubenson et al., 2012 JEB).

Additional comments

This is a nice manuscript- it is interesting and well written. That incline running should exact an additional metabolic cost not only due to the work required for raising the center of mass uphill is an important concept and a valuable contribution to the literature. Nice work!

---

## Round 0.2 · Minor Revisions

The reviewers are generally happy with the revision, but a few minor points need to be addressed before the paper is accepted for publication.

·

Basic reporting

No comments

Experimental design

No comments

Validity of the findings

No comments

Additional comments

The authors have done a good job in revising the manuscript and I think it is considerably improved in this new version.

GENERAL COMMENTS:
One of the motivations for the study is that earlier models on the energy cost of uphill running produce unrealistic values for efficiency. It is still not clear to me exactly on what basis these values are being deemed unrealistic. For example on Lines 48 and 55, the efficiencies from previous models are referred to as "low" and "high", respectively. Are these efficiencies too low and too high because they are very different from what muscle physiology tells us the efficiency of a stretch-shortening cycle or pure concentric work is? I think it would help to clarify in the text why these values are unrealistic.

I still feel like there are too many non-specific terms used in the Introduction that are left up to the reader’s interpretation or requires very careful reading to figure out what specific definitions of these terms the authors are actually using. For example:

Lines 34-40: “There is no net mechanical power required” I don't think this statement is accurate; there are definitely periods of non-zero mechanical power during the gait cycle. Do the authors mean there is no net mechanical work required? Even that is only true for certain definitions of mechanical work (Sasaki et al., 2009). For example the net joint work over a level gait cycle is not zero (Devita et al., 2007). I think the authors are referring to the “external” work/power here in the tradition of Cavagna; if so, it would be helpful for clarity to use this term explicitly (external work) and cite their work (Cavagna et al., 1963, 1964) the first time the term is used.

Line 45: Cycle ergometry is mostly concentric work but it is not devoid of eccentric work (e.g. Neptune & van den Bogert, 1998). I would suggest also citing a reference from the muscle physiology literature (e.g. Smith et al., 2005; their Table 2 notes a “net efficiency” (work/enthalpy) of 27% for concentric work at an "optimal" (slow) shortening velocity).

In response to my question in the first review on the definition of “perpendicular bouncing”, the authors replied:

"Conceptually, you are correct. But as stated, to distinguish between supporting body weight and lifting the COM vertically (increasing the total potential energy), we considered body weight support to be perpendicular to the surface. We think this consideration has minor consequences, since for a 9 degree angle the perpendicular component is only slightly less than the vertical component: cosine 9 degrees equals 0.988."

I suggest adding this description in the text of the paper. If it is in there already in the new version, I missed it. I think readers will be very confused by using a direction other than absolutely vertical to refer to lifting the CoM vertically.

SPECIFIC (MINOR) COMMENTS:
Lines 13-15: I’m not sure if this statement is necessarily true (that we have an objectively good biomechanical explanation for the energetic cost of level running), but some more references could probably be added here (e.g. Högberg, 1952; Williams & Cavanagh, 1987; Gutmann et al., 2006).

Line 33: Is the running velocity here the horizontal velocity or the velocity parallel to the incline?

Line 49: I think “Delta Efficiency” (change in work over change in energy) is the more common term used for this (rather than vertical efficiency).

Line 74: I would remove “shorten and” from here. There is a great deal of eccentric muscle work in gait, by any definition, even when running uphill.

REFERENCES
Cavagna GA, Saibene FP, and Margaria R (1963). External work in walking. Journal of Applied Physiology 18, 1-9.

Cavagna GA, Saibene FP, and Margaria R (1964). Mechanical work in running. Journal of Applied Physiology 19, 249-256.

DeVita P, Helseth J, and Hortobagyi T (2007). Muscles do more positive than negative work in human locomotion. Journal of Experimental Biology 210, 3361-3373.

Gutmann AK, Jacobi B, Butcher MT, and Bertram JEA (2006). Constrained optimization in human running. Journal of Experimental Biology 209, 622-632.

Högberg P (1952). How do stride length and frequency influence the energy-output during running. Internationale Zeitschrift für Angewandte Physiologie 14, 437-441.

Neptune RR and van den Bogert AJ (1998). Standard mechanical energy analyses do not correlate with muscle work in cycling. Journal of Biomechanics 31, 239-245.

Sasaki K, Neptune RR, and Kautz SA (2009). The relationships between muscle, external, internal and joint mechanical work during normal walking. Journal of Experimental Biology 212, 738-744.

Smith NP, Barclay CJ, and Loiselle DS (2005). The efficiency of muscle contraction. Progress in Biophysics & Molecular Biology 88, 1-58.

Williams KR and Cavanagh PR (1987). Relationship between distance running mechanics, running economy, and performance. Journal of Applied Physiology 63, 1236-1245.

·

Basic reporting

In terms of the basic reporting, I think the manuscript still reads well and the authors have addressed my previous comments in this area. I have a couple of minor comments that hopefully provide useful feedback on the clarity of some terminology that I found awkward:

Figure 1 now nicely spells out the components of the model but should the cost of perpendicular bouncing be included in the cost of parallel running? It's really just an issue of terminology but it seems non-intuitive for a cost of motion that is perpendicular to the surface to come under the parallel costs. Perhaps the net metabolic cost of running should be split into three sub-costs based on the three terms in the model [i.e. 1. cost of power for raising the COM; 2. cost of parallel running (braking and propulsion costs); 3. Cost of perpendicular bouncing + cost of swinging the limbs].

Your use of the term net propelling impulse is somewhat confusing. I immediately think of net impulse as being the sum of total braking and propulsive impulses that equals zero for constant velocity locomotion. I appreciate that you define this term but I personally found it confusing and suggest changing it for clarity.

Again, purely a personal opinion, but I also find the term wasted work a bit misleading. I understand that you are being consistent with previous work but there is considerable potential for storage and return of negative work in elastic structures. The term 'wasted work' implies that all the negative work and subsequent positive work is done actively by muscle, which is not likely to be the case.

Experimental design

No Comments

Validity of the findings

No Comments

Additional comments

I think the changes made by the authors have strengthened the manuscript and appreciate the lengths they have gone to in order to respond to the initial review. The additional data has provided good justification for the use of an exponential fit for the decline in braking and propulsion, which was my biggest concern with the original paper. I think this work will make an excellent contribution to the field.

·

Basic reporting

The study "Applying the cost of generating force hypothesis to uphill running" offers a novel model for the determinant of the metabolic cost of incline running. This is a nice addition to the field or locomotor mechanics/energetics.

The authors have done an excellent job at addressing all of the reviewer comments. In particular, they have gone to lengths to collect new data that improves the quality of the study.

Nice Study!

I only have two very minor comments:

line 51, "no effect" - no effect on what? the mechanics of running? The net work of running?

line 116-117, is this statement a bit strong?

Experimental design

No Comments

Validity of the findings

No Comments

Additional comments

No Comments

---

## Round 0.3 · accepted · Accept

Thanks for your revisions. The reviewers are happy with the manuscript in its present form.

·

Basic reporting

I have no further comments

Experimental design

I have no further comments

Validity of the findings

I have no further comments

Additional comments

Thank you for responding to my few remaining comments. I have no further comments to make and am happy for the manuscript to be published.

·

Basic reporting

All reviewer comments have been addressed well. This study is a nice addition to the field.

Experimental design

No comments

Validity of the findings

No comments

Additional comments

No comments